# The Hematopoietic Effect of Ninjinyoeito (TJ-108), a Traditional Japanese Herbal Medicine, in Pregnant Women Preparing for Autologous Blood Storage

**DOI:** 10.3390/medicina58081083

**Published:** 2022-08-11

**Authors:** Eriko Fukuda, Takuya Misugi, Kohei Kitada, Megumi Fudaba, Yasushi Kurihara, Mie Tahara, Akihiro Hamuro, Akemi Nakano, Masayasu Koyama, Daisuke Tachibana

**Affiliations:** Department of Obstetrics and Gynecology, Graduate School of Medicine, Osaka Metropolitan University, 1-4-3 Asahimachi Abeno-ku Osaka, Osaka 545-8585, Japan

**Keywords:** Ninjinyoeito, autologous blood storage, postpartum hemorrhage, hematopoiesis, pregnancy, placenta previa

## Abstract

*Background and Objectives:* There are no reports showing the hematopoietic effect of TJ-108 on pregnant women. The aim of this study was to investigate the effect of TJ-108 on the hemoglobin and hematocrit levels, and white blood cell and platelet counts of pregnant women complicated with placenta previa who were managed with autologous blood storage for cesarean section. *Materials and Methods:* We studied two groups of patients who were complicated with placenta previa and who underwent cesarean delivery. Group A consisted of women who were treated with oral iron medication (100 mg/day), and Group B consisted of women who were treated with TJ-108 at a dose of 9.0 g per day, in addition to oral iron medication, from the first day of blood storage until the day before cesarean delivery. To evaluate the effect of TJ-108, the patients’ red blood cell (RBC); Hb; hematocrit (Ht); white blood cell (WBC); and platelet count (PLT) levels were measured 7 days after storage and at postoperative days (POD) 1 and 5. *Results:* The study included 65 individuals, 38 in group A and 27 in group B. At the initial storage, a 0.2 g/dL reduction in Hb levels was observed, as compared to the initial Hb levels, in the TJ-108 treated patients, whereas a 0.6 g/dL reduction in Hb levels was observed in the non-TJ-108 treated group. On the other hand, regarding the second and subsequent storages, no significant difference was found in the decrease in the Hb levels of both groups. *Conclusions:* This study is the first report showing the effect of TJ-108 on improving anemia in pregnant women, presumably by its boosting effect on myelohematopoiesis. Therefore, the combined administration of both iron and TJ-108 is effective as a strategy for pregnant women at a high risk of PPH due to complications such as placenta previa.

## 1. Introduction

Postpartum hemorrhage (PPH) is one of the leading causes of maternal death. Multidisciplinary treatment, including blood transfusion, should be set up beforehand, especially for pregnant women at a high risk of PPH due to such conditions as placenta previa or placenta accreta [1]. Hemorrhage during cesarean section that is associated with placenta previa is highly correlated with the need for allogenic blood transfusion [2]. Despite the markedly improved safety of allogenic blood transfusion, the risk of transmission of viral or bacterial infection cannot be completely eliminated [3]. Placenta previa is one of the most serious complications during pregnancy and is associated with increased blood loss at delivery; it is also an important cause of serious fetal and maternal morbidity and mortality [4]. Prenatal diagnosis, followed by the careful planning of cesarean delivery and preparation for possible blood loss by a multidisciplinary team, reduces the risk of fetal and maternal morbidity and mortality [5]. Allogeneic blood transfusion has been used for postpartum hemorrhage, although there are substantial risks, such as viral infection, allergy, posttransfusion immune suppression and graft versus host disease [6]. The usefulness of autologous blood transfusion has been previously reported [7,8,9]; however, this method of blood storage can lead to preoperative anemia.

As an alternative to allogenic blood transfusion, preoperative autologous blood storage is useful [10], although the process can induce anemia.

Ninjinyoeito (TJ-108) (Tsumura Co., Tokyo, Japan) is a Japanese herbal medicine that has been used to improve anemia in patients for several years [11]. TJ-108 is composed of 12 unrefined ingredients in fixed proportion: 3.0 g of ginseng; 4.0 g of Japanese angelica root; 2.0 g of peony root; 4.0 g of rehmannia root; 4.0 g of atractylodes rhizome; 4.0 g of poria sclerotium; 2.5 g of cinnamon bark; 1.5 g of astragalus root; 1.5 g of unshiu peel; 2.0 g of polygala root; 1.0 g of schisandra fruit; and 1.0 g of glycyrrhiza. However, to our knowledge, there have been no reports showing the hematopoietic effect of TJ-108 on pregnant women until now.

The aim of this study was to investigate the effect of TJ-108 on the hemoglobin and hematocrit levels, and white blood cell and platelet counts of pregnant women complicated with placenta previa who were managed with autologous blood storage for cesarean section.

## 2. Materials and Methods

This retrospective observational study was approved by the institutional review board (Approved Number: 2022-030, 13 June 2022). Between January 2016 and December 2020, pregnant women who underwent cesarean delivery with the indication of placenta previa were selected for this study. Patients whose hemoglobin (Hb) level was less than 10.0 g/dL were excluded from this study. Cesarean delivery was performed at 36 or 37 gestational weeks. Data of each patient were retrospectively obtained from medical records.

Autologous blood storage was performed when the patient’s Hb level was above 10.0 g/dL and canceled at less than 10.0 g/dL. All patients had 300 mL of blood stored each time for a maximum of 1200 mL (4 times). The initial storage was scheduled according to the gestational weeks. The storage was scheduled every week until 7 days before the scheduled day of cesarean delivery. All patients took a daily oral iron medication (100 mg per day) from the day of the initial storage until the day before the cesarean delivery. From 2019, TJ-108 prescriptions in addition to iron administration were initiated for all women who performed autologous blood storage.

We studied two groups of patients. Group A consisted of women who underwent cesarean delivery between January 2016 and December 2018 who did not take TJ-108. Group B consisted of women who underwent cesarean delivery between January 2019 and December 2020 who were treated with TJ-108 at a dose of 9.0 g per day, in addition to the oral iron medication, from the first day of blood storage until the day before the cesarean delivery. Informed consent for the research was obtained from all patients. We informed the patients of their choice to opt-out.

To evaluate the effect of TJ-108, red blood cell (RBC); Hb; hematocrit (Ht); white blood cell (WBC); and platelet count (PLT) levels were measured after 7 days of storage and at postoperative days (POD) 1 and 5. For laboratory tests, serum aspartate aminotransferase (AST); alanine aminotransferase (ALT); creatinine; blood urea nitrogen; sodium; potassium; and chlorine were measured on the day of storage and at POD 1.

Cesarean delivery was performed on the scheduled day for non-eventful patients, and emergency cesarean delivery was performed in cases with bleeding or uterine contraction. All patients were allowed to take clear liquid until 3 h before the scheduled operation and were administered a continuous infusion of lactate Ringers’ solution (200 mL/h). The intraoperative loss of blood (including amniotic fluid) was estimated by measuring the amount of blood in the suction collection unit and by weighting the used surgical gauzes.

Continuous variables were expressed as median (range), and categorical variables were expressed as a number (%). Statistical analyses were examined using the Mann–Whitney U-test and Fisher’s exact test performed with BellCurve for Excel (Social Survey Research Information Co., Ltd., Tokyo, Japan). A *p*-value of < 0.05 was considered to indicate significance.

## 3. Results

During the survey period, 38 patients were treated without TJ-108 (Group A), and 27 patients were treated with TJ-108 (Group B). Table 1 shows the clinical characteristics of the participants in both groups, and the characteristics were not significantly different between the two.

The characteristics of maternal and neonatal outcomes data are shown in Table 2. Allogeneic transfusion during cesarean delivery was not significantly different between the two groups, and there was no significant difference between the two in the total amount of blood storage. The median autologous transfusion was 300 mL in both groups, and the frequency of transfusion during cesarean section or blood loss wase not significantly different between the two groups.

Table 3 shows the results of the patients’ complete blood count before and 7 days after the initial blood storage of both groups. Hb levels were significantly higher in Group B at 7 days after the initial storage (Group A: 10.0 g/dL; Group B: 10.6 g/dL. *p* = 0.001, 95% confidence interval, 0.13–1.01) (Figure 1a,b). Moreover, the Hb level in Group B was reduced by 0.2 g/dL, as compared to the initial Hb level before storage, whereas the Hb level was reduced by 0.6 g/dL in Group A (*p* = 0.012). The cancellation rate of blood storage in the next instance was significantly lower in group B due to the reduced decrease in hemoglobin levels.

Table 4 shows the results of the parameters after the second and subsequent storages in both groups. There were no significant differences in Hb and Ht levels (Hb: 10.7 vs. 10.5; *p* = 1.000, Ht: 32.2 vs. 31.8; *p* = 0.259, respectively) (Figure 1c,d).

No significant differences were found between the two groups regarding white blood cell and platelet levels after 7 days of blood storage, and regarding Hb, Ht and RBC on POD 1 and 5. Blood tests on POD 1 showed elevated AST and ALT in two cases in group A and two cases in group B. No other abnormalities were found in other laboratory values. 

## 4. Discussion

The present study first revealed that the reduction in Hb levels in the TJ-108 treated patients was significantly less than that observed in the non-TJ-108 treated patients 7 days after storage. At the initial storage, a 0.2 g/dL reduction in Hb levels was observed, as compared to the initial Hb levels in the TJ-108 treated patients, whereas a 0.6 g/dL reduction in Hb levels was observed in the non-TJ-108 treated group. On the other hand, regarding the second and subsequent storages, no significant difference was found in the decrease in Hb levels of both groups.

Takano et al. reported that an oral administration of TJ-108 protected against hematotoxicity in mice treated with 5-fluorouracil (5-FU), which causes severe anemia [12]. They showed that TJ-108 inhibited 5-FU-induced decreases in peripheral reticulocyte and bone marrow cell counts on day 10, and markedly hastened their recovery on day 20 in a dose-dependent manner. Erythroid progenitor colonies, such as colony forming units-erythroid (CFU-E) and burst forming units-erythroid (BFU-E) formed by marrow cells from mice treated with 5-FU were significantly increased by an oral administration of TJ-108 [13].

In the differentiation of erythroblastic cells, BFU-E becomes CFU-E cells, and they then differentiate into erythroblasts. During the late stage of erythroid differentiation, proerythroblasts undergo mitosis to produce basophilic, polychromatic and orthochromatic erythroblasts, and these orthochromatic erythroblasts expel their nuclei to generate reticulocytes. Finally, the reticulocytes mature into RBC, initially in bone marrow, and then in the circulation [14]. In these processes of RBC differentiation, EFU-E or CFU-E are in an earlier stage of differentiation dependent on erythropoietin, and erythroblast or normoblasts are in a later stage of differentiation dependent on iron. [15]. In our study, TJ-108 showed a significant potentiate effect in hematogenesis at the initial storage, yet it had no effect at the second and subsequent blood storages. This may be because TJ-108 stimulated BFU-E, CFU-E or the upper stream of differentiation of erythroblasts and showed a synergistically boosting effect on myelohematopoiesis with iron preparation. Promoting hematopoiesis after blood storage takes some time, and the TJ-108-treated group showed an early effect due to boosting. Therefore, it showed a difference from the non-treated group. However, after the second and subsequent storage, the hematopoiesis of the non-treated group caught up, and no difference was observed.

Hatano et al. reported that angelica roots, one of the components of TJ-108, increase the recovery of erythrocytopenia and stimulates the differentiation of erythroid progenitors without promoting erythropoietin synthesis. They also reported that TJ-108 lowered plasma interferon-γ levels, which may suppress the activity of erythroid progenitor cells. They considered the possibility that the polysaccharides in angelica roots promote hematopoiesis by activating immature erythroid cells, in part, by suppressing cytokine secretion [16].

Presently, complementary and alternative medicines, such as traditional Japanese Kampo medicines, are frequently used together with Western medicines for the treatment of various diseases, including anemia. Motoo et al. showed a randomized controlled trial with TJ-108 for patients complicated with hepatitis C and receiving ribavirin, which is known to cause severe anemia. They reported that a maximal decrease in Hb in the TJ-108 group was significantly smaller than that in the control group (TJ-108: 2.59 g/dL vs. non-TJ-108: 3.71 g/dL, respectively). They also concluded that TJ-108 could be used as a supportive remedy to reduce the ribavirin-induced anemia in the treatment of chronic hepatitis C [13].

In Japan, the current health insurance system covers the prescription of Kampo medicines including TJ-108, available as both herbs for decoctions and extract formulations. It has been reported that herbal medicines are widely used worldwide to treat a variety of ailments during pregnancy [17,18,19]. Though there have been no reports of TJ-108 being administered to pregnant women, the herbal medicines that comprise TJ-108 have been reported to be safe for use in pregnant women [20].

There were some limitations to this study. Firstly, as we did not evaluate the condition of iron or related inspection items (reticulocytes, ferritin, transferrin saturation, vitamin B12 and folates), we were unable to determine the effect of TJ-108 on bone marrow’s ability to produce new blood cells. Secondly, we retrospectively reviewed a relatively small number of patients. Further studies are needed to elucidate the effect of TJ-108 on bone marrow activity in autologous blood storage.

## 5. Conclusions

This study is the first report showing the effect of TJ-108 on improving anemia in pregnant women, presumably with a boosting effect on myelohematopoiesis. It is suggested that the combined administration of iron and TJ-108 is an effective strategy for pregnant women at a high risk of PPH such as placenta previa.

## Figures and Tables

**Figure 1 medicina-58-01083-f001:**
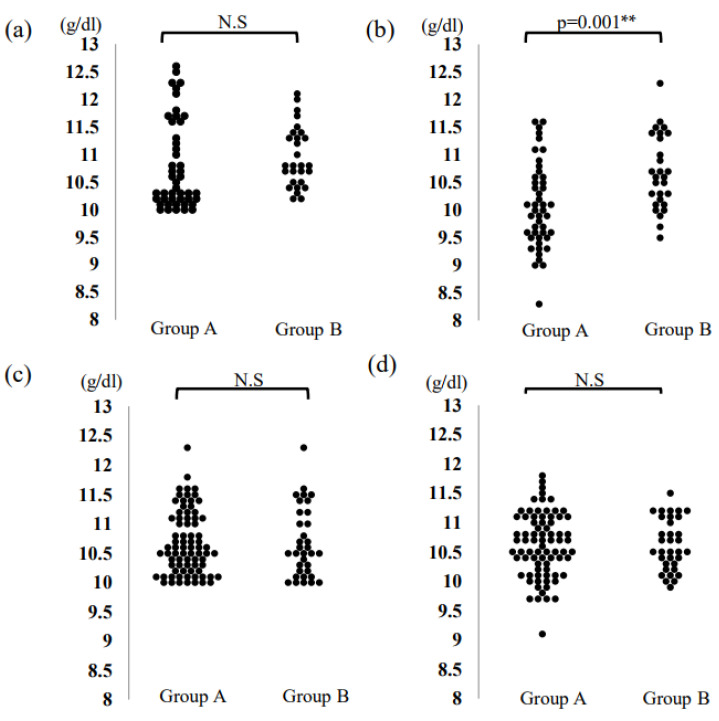
The dot-plot diagram shows the distribution of hemoglobin values (**a**) before the initial storage; (**b**) seven days after the initial storage; (**c**) before the second or subsequent storage; (**d**) seven days after the second or subsequent storage. ** *p* < 0.01, N.S: not significant.

**Table 1 medicina-58-01083-t001:** Clinical characteristics of participants.

Variables	Group A	Group B	*p* Value
	*n* = 38	*n* = 27	
	Number or Median (Range)	Number or Median (Range)	
Age (year)	35 (28–44)	34 (24–44)	0.310
Height (cm)	160 (140–165)	160 (151–169)	0.329
Body weight before pregnancy (kg)	52 (41–65)	51 (44–87)	0.669
BMI at before pregnancy (kg/m^2^)	20.7 (16.4–27.8)	20.4 (17.5–32.7)	0.926
Body weight at birth (kg)	61 (47–72)	59 (50–89)	0.680
BMI at birth (kg/m^2^)	24 (21.0–28.5)	24.2 (19.3–33.5)	0.963
Gestational age (week)	37.0 (34.4–37.3)	37 (32.4–37.7)	0.942
Primigravida (%)	15 (39.5)	15 (55.6)	0.200
ART (%)	6 (15.8)	7 (25.9)	0.314
Emergency cesarean delivery (%)	7 (18.4)	6 (22.2)	0.706
Apgar score 1 min	8 (1–9)	8 (3–9)	0.660
Apgar score 5 min	9 (5–9)	9 (6–9)	0.225
Birth weight (g)	2643 (1966–3440)	2500 (1917–3035)	0.250
Male (%)	24 (63.2)	11 (40.7)	0.074
Female (%)	14 (36.8)	16 (59.3)	0.074

BMI: body mass index; ART: assisted reproductive technology.

**Table 2 medicina-58-01083-t002:** Operative outcomes of both groups.

Variables		Group A	Group B	*p* Value
		*n* = 38	*n* = 27	
		Number or Median (Range)	Number or Median (Range)	
Total of autologous blood storage (mL)		600 (300–1200)	600 (300–1200)	0.589
Blood transfusion during cesarean section	Autologous transfusion (mL)	300 (0–1200)	300 (0–1200)	0.336
	Allogeneic transfusion (%)	8 (21.1)	8 (29.7)	0.121
	RBC (unit)	0 (0–10)	0 (0–8)	0.729
	FFP (unit)	0 (0–10)	0 (0–6)	0.779
	PC (unit)	0 (0–10)	0 (0–0)	0.919
Infusion (mL)		1950 (850–4300)	1450 (700–3400)	0.186
Blood loss (mL)		1835 (340–7500)	1760 (895–5500)	0.863
Urine output (mL)		110 (0–550)	80 (0–500)	0.105
Operation time (min)		65 (39–148)	68 (41–104)	0.739

RBC: red blood cell; FFP: fresh frozen plasma; PC: platelet concentrates.

**Table 3 medicina-58-01083-t003:** Blood cell counts and differences in the initial autologous blood storage.

	Group A	Group B	*p* Value
	38 Storage (*n* = 38)	27 Storage (*n* = 27)	
	Median (Range)	Median (Range)	
Cancellation rate of next storage (%)	17 (44.7)	3 (11.1)	0.004 *
Hb level before storage (g/dL)	10.5 (10.0–12.6)	10.8 (10.2–12.1)	0.070
Hb level after 7 days of storage (g/dL)	10.0 (8.3–11.6)	10.6 (9.5–12.3)	0.001 **
Amount of change in Hb (g/dL)	−0.6 (−2.1–0.4)	−0.2 (−1.5–0.6)	0.012 *
Ht level before storage (%)	31.9 (29.6–39.5)	32.3 (29.8–36.3)	0.394
Ht level after 7 days of storage (%)	30.5 (27.7–36.5)	32.1 (28.4–36.4)	0.015 *
Amount of change in Ht (%)	−1.8 (−5.8–1.9)	−0.7 (−2.5–1.2)	0.015 *
RBC level before storage (×10^4^/μL)	356 (296–433)	354 (318–430)	0.730
RBC level after 7 days of storage (×10^4^/μL)	333 (270–396)	349 (299–398)	0.125
Amount of change in RBC (×10^4^/μL)	−24 (−69–12)	−11.5 (−45–12)	0.003 **

Hb: hemoglobin; Ht: hematocrit; RBC: red blood cell. * *p* < 0.05, ** *p* < 0.0.

**Table 4 medicina-58-01083-t004:** Blood cell counts and differences in the second and subsequent storages.

	Group A	Group B	*p* Value
	55 Storage (*n* = 38)	38 Storage (*n* = 27)	
	Median (Range)	Median (Range)	
Cancel rate of next storage (%)	6 (10.9)	1 (2.7)	0.137
Hb level before storage (g/dL)	10.6 (10.0–11.8)	10.5 (10.0–12.3)	0.769
Hb level after 7 days of storage (g/dL)	10.7 (9.1–11.8)	10.5 (9.9–11.5)	1.000
Amount of change in Hb (g/dL)	−0.1 (−1.0–1.1)	0.0 (−1.4–0.6)	0.692
Ht level before storage (%)	32.2 (29.0–37.2)	31.8 (28.9–36.4)	0.367
Ht level after 7 days of storage (%)	32.2 (28.8–37.2)	31.7 (28.6–34.1)	0.259
Amount of change in Ht (%)	0.2 (−3.4–3.2)	−0.1 (−3.1–2.7)	0.629
RBC level before storage (×10^4^/μL)	343 (305–404)	337 (298–398)	0.205
RBC level after 7 days of storage (×10^4^/μL)	344 (292–392)	332 (297–384)	0.082
Amount of change in RBC (×10^4^/μL)	−4 (−36–24)	−8 (−34–18)	0.841

Hb: hemoglobin; Ht: hematocrit; RBC: red blood cell.

## Data Availability

The datasets used and/or analyzed during the current study are available from the corresponding author on reasonable request.

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
