# Peer review of "The Hematopoietic Effect of Ninjinyoeito (TJ-108), a Traditional Japanese Herbal Medicine, in Pregnant Women Preparing for Autologous Blood Storage"

_medicina, 2022, doi:10.3390/medicina58081083_

Round 1

Reviewer 1 Report

This study evaluated the hematopoietic effect of TJ-108 (Ninjin-yoei-to). The authors compared hemoglobin and hematocrit levels between the non-treated and TJ-108-treated groups of high-risk PPH patients with autologous blood storage. In the TJ-108-treated group, the authors showed higher Hb levels and a smaller decrease seven days after the first blood storage compared to the non-treated group. The authors suggested that TJ-108 has a boosting effect on myelohematopoiesis and contributes to improving anemia in pregnant women who experience blood storage.

In clinical practice, blood storage of PPH high-risk patients is crucial. Unfortunately, we have sometimes experienced that we can store only a small amount of autologous blood storage due to low Hb levels. Therefore, this study is of potential clinical interest. However, the significance of what is demonstrated in this study is minor and should make a more clinically relevant suggestion for publication. Moreover, the text is inadequate in many respects.

1. This study investigated the effect of TJ-108 on hemoglobin and hematocrit levels in pregnant women. However, because the authors concluded that TJ-108 is an effective strategy for high-risk PPH pregnant women, the direct clinical benefit (e.g., canceled rate of storage, the amount of allogeneic transfusion) of using TJ-108 should be demonstrated, not just the improvement in laboratory values.

2. The main result was the amount of change in Hb or Ht or RBC level. It isn't easy to focus on the results because table 3 and 4 contains too many variables. I recommend visualizing the most impressive results in a dot plot. Also, what are the implications of the WBC and PLT values? Please consider removing these data from the table for better visibility.

Abstract:

3. Line 28: "myelohematopoiecis" is incorrect. It should be replaced by "myelohematopoiesis."

Material and Methods:

4. Line 60: "theis" is incorrect. It should be replaced by "this."

5. Line 71–72: Why did clinicians start adding TJ-108 treatment in January 2019? Is TJ-108 used empirically in pregnant women? Please describe the clinical or scientific reason for initiating TJ-108 treatment.

6. Line 74: This was a retrospective study. Please confirm that the authors have obtained informed consent from all patients or just informed them of the option to opt-out.

7. Line 77: Authors did not describe the data of POD1 and 5 in the Results section.

8. Line 78–79: Authors did not describe the data of serum aspartate aminotransferase, alanine aminotransferase, creatinine, blood urea nitrogen, sodium, potassium and chlorine in the result section.

9. Line 87–89: Why did the Authors express all continuous variables as median and range instead of the mean and standard deviation? (Also, the Mann-Whitney U-test was used for statistical analysis instead of the Student's t-test.)  A similar previous study used mean+/-SD and t-test.

YAMAMOTO, Yasuhiro, et al. Safety and efficacy of preoperative autologous blood donation for highrisk pregnant women: Experience of a large university hospital in Japan. Journal of Obstetrics and Gynaecology Research, 2014, 40.5: 1308–1316.

Results:

10. Line 124: "Table 4. This is a table. Tables should be placed in the main text near to the first time they are cited." The appropriate table title should replace.

11: Table 4: Font sizes should be consistent.

Discussion:

12. Line 128–130, Line 135–136, Line 136–137: The author repeats text that means the same thing.

13. Line 138–146: The authors should move the description of placenta previa and autologous blood transfusion to the introduction section. The authors should include a discussion based on the results in this section.

14. Line 166–170: It isn't easy to understand why TJ-108 did not show significant hematopoietic effects after the second and subsequent storage. My understanding is as follows:

Promoting hematopoiesis after blood storage takes some time, and the TJ-108-treated group showed an early effect due to boosting. Therefore, it showed a difference from the non-treated group. However, after the second and subsequent storage, the hematopoiesis of the non-treated group caught up, and no difference was observed.

Is this correct?

15. Line 192–195: Authors should remove the template text.

16. Authors should describe the safety of TJ-108 for pregnant women.

Author Response

  1. This study investigated the effect of TJ-108 on hemoglobin and hematocrit levels in pregnant women. However, because the authors concluded that TJ-108 is an effective strategy for high-risk PPH pregnant women, the direct clinical benefit (e.g., canceled rate of storage, the amount of allogeneic transfusion) of using TJ-108 should be demonstrated, not just the improvement in laboratory values.

→We have added the cancellation rate of next storage in the Table 3 and 4.

The following text was added.

“The cancellation rate of blood storage in the next chance was significantly lower in group B due to the reduced decrease in hemoglobin levels.”

The usage rate of allogeneic transfusion was described in Table 2.

  1. The main result was the amount of change in Hb or Ht or RBC level. It isn't easy to focus on the results because table 3 and 4 contains too many variables. I recommend visualizing the most impressive results in a dot plot. Also, what are the implications of the WBC and PLT values? Please consider removing these data from the table for better visibility.

→We have added the dot plot diagram of distribution of hemoglobin values. We have removed the data of WBC and PLT values.

Abstract:

  1. Line 28: "myelohematopoiecis" is incorrect. It should be replaced by "myelohematopoiesis."

→We have revised.

Material and Methods:

  1. Line 60: "theis" is incorrect. It should be replaced by "this."

→We have revised.

  1. Line 71–72: Why did clinicians start adding TJ-108 treatment in January 2019? Is TJ-108 used empirically in pregnant women? Please describe the clinical or scientific reason for initiating TJ-108 treatment.

The following text was added.

“Since 2019, TJ-108 prescriptions in addition to iron administration were initiated for all women who performed autologous blood storage.”

  1. Line 74: This was a retrospective study. Please confirm that the authors have obtained informed consent from all patients or just informed them of the option to opt-out.

→We just informed the patients the option to opt-out, so we changed to the following sentence.

“We informed the patients of the option to opt-out.”

  1. Line 77: Authors did not describe the data of POD1 and 5 in the Results section.

→We also examined the blood test results on day 1 and 5, but removed them because there was no difference between the two groups and they were not the primary focus of this study.

  1. Line 78–79: Authors did not describe the data of serum aspartate aminotransferase, alanine aminotransferase, creatinine, blood urea nitrogen, sodium, potassium and chlorine in the result section.

→The following sentences were added.

“Blood tests on the POD 1 showed elevated AST and ALT in 2 cases in group A and 2 cases in group B. No other abnormalities were found in other laboratory values.”

  1. Line 87–89: Why did the Authors express all continuous variables as median and range instead of the mean and standard deviation? (Also, the Mann-Whitney U-test was used for statistical analysis instead of the Student's t-test.) A similar previous study used mean+/-SD and t-test.

YAMAMOTO, Yasuhiro, et al. Safety and efficacy of preoperative autologous blood donation for high‐risk pregnant women: Experience of a large university hospital in Japan. Journal of Obstetrics and Gynaecology Research, 2014, 40.5: 1308–1316.

→We did not use Student's t tests because most of the data were not normally distributed thus using Student's T test can produce biased results.

Results:

  1. Line 124: "Table 4. This is a table. Tables should be placed in the main text near to the first time they are cited." The appropriate table title should replace.

→The appropriate table title was replaced as follows.

“Blood cell counts and differences in the second and subsequent storages.”

11: Table 4: Font sizes should be consistent.

→Font size was unified to 10.

Discussion:

  1. Line 128–130, Line 135–136, Line 136–137: The author repeats text that means the same thing.

→The following sentences were deleted.

“This is the first report to show the hematopoietic effect of TJ-108 on pregnant women.  We firstly revealed that a combination use of TJ-108 with iron had a significant boosting effect on improving anemia in pregnant women at high risk of PPH.”

  1. Line 138–146: The authors should move the description of placenta previa and autologous blood transfusion to the introduction section. The authors should include a discussion based on the results in this section.

→These sentences were deleted and incorporated into the introduction.

“Placenta previa is one of the most serious complications during pregnancy and is associ-ated with increased blood loss at delivery; it is also an important cause of serious fetal and maternal morbidity and mortality [6]. Prenatal diagnosis, followed by the careful plan-ning of cesarean delivery and preparation for possible blood loss by a multidisciplinary team reduces the risk off fetal and maternal morbidity and mortality [7]. Allogeneic blood transfusion has been used for postpartum hemorrhage, although there are substantial risks, such as viral infection, allergy, posttransfusion immune suppression or and graft- versus- host disease [8]. The usefulness of autologous blood transfusion has been previously reported [9-12]; however, this method of blood storage can lead to preoperative anemia.”

  1. Line 166–170: It isn't easy to understand why TJ-108 did not show significant hematopoietic effects after the second and subsequent storage. My understanding is as follows:

Promoting hematopoiesis after blood storage takes some time, and the TJ-108-treated group showed an early effect due to boosting. Therefore, it showed a difference from the non-treated group. However, after the second and subsequent storage, the hematopoiesis of the non-treated group caught up, and no difference was observed.

Is this correct?

→Thank you for pointing out this important point. We inserted the following sentences.

“Promoting hematopoiesis after blood storage takes some time, and the TJ-108-treated group showed an early effect due to boosting. Therefore, it showed a difference from the non-treated group. However, after the second and subsequent storage, the hematopoiesis of the non-treated group caught up, and no difference was observed.”

  1. Line 192–195: Authors should remove the template text.

→We removed.

  1. Authors should describe the safety of TJ-108 for pregnant women.

→The following sentences were deleted.

“In Japan, the current health insurance system covers prescription of Kampo medicines including TJ-108, available as both herbs for decoctions and extract formulations. It has been reported that herbal medicines are widely used worldwide to treat a variety of ailments during pregnancy [21-23]. Though there have been no reports of TJ-108 being administered to pregnant women, the herbal medicines that comprise TJ-108 have been reported to be safe for use in pregnant women [24]. ”

Reviewer 2 Report

The authors of the present study investigated the effect of TJ-108, a traditional Japanese herbal medicine in combination with an oral iron preparation on blood count parameters in pregnant women donating autologous blood because of a presumed future blood loss due to placenta previa.

Apart from the fact that autologous blood donation is not feasible in a high number of patients with placenta previa (see cited reference #9) and the use of autologous blood transfusion in general has decreased significantly in recent years, the quality of the manuscript may be increased.

It should be noted in the abstract, that both treatment groups (including group A) received po iron 100 mg/d. 

Additional information about iron status (at least ferritin and MCV should have been routinely assessed in pregnant women) as well as reticulocyte count/transferrin saturation and B12/folates may be useful to better compare the two cohorts.

It may be discussed that the small number of investigated cases may not be sufficient for final conclusions.

Author Response

Apart from the fact that autologous blood donation is not feasible in a high number of patients with placenta previa (see cited reference #9) and the use of autologous blood transfusion in general has decreased significantly in recent years, the quality of the manuscript may be increased.

→Reference 9 is not appropriate for this issue and has been removed.

It should be noted in the abstract, that both treatment groups (including group A) received po iron 100 mg/d.

→We inserted the following text.

“who were treated with oral iron medication (100mg/day)”

Additional information about iron status (at least ferritin and MCV should have been routinely assessed in pregnant women) as well as reticulocyte count/transferrin saturation and B12/folates may be useful to better compare the two cohorts.

→Unfortunately, we did not evaluate these items. We inserted the following text in the limitation.

“as we did not evaluate the condition of iron or related inspection items (reticulocytes, ferritin, transferrin saturation, Vitamin 12 and folates),”

It may be discussed that the small number of investigated cases may not be sufficient for final conclusions.

→We inserted the following text.

“, so it may not be sufficient for final conclusions.”

Round 2

Reviewer 1 Report

The authors have responded appropriately to my comments. 

1. The Discussion section is too long in the description of TJ-108. Consider moving some sentences to the Introduction section or removing them.

2. Authors should modify the conclusion. Not add the text "so it may not be sufficient for final conclusions" (Line 209), but modified the conclusion consistent with the study aim and result. The current author's conclusion "The combined administration of iron and TJ-108 is an effective strategy for pregnant women at high risks of PPH such as placenta previa." can not be affirmed because of the small sample number and the authors did not show sufficient direct clinical benefit. How about using the expression "suggested"?

Author Response

We appreciate your kind advice and thoughtful comments on refining our manuscript.

1. The Discussion section is too long in the description of TJ-108. Consider moving some sentences to the Introduction section or removing them.

→We removed the followed sentence because it seems to have little to do with the content of this study. Also, we remove the reference [12, 13, 14, 15, 16].

TJ-108 has multi-functional, beneficial activities and is commonly used for the improvement of recovery from diseases or symptoms, such as neurodisorders [12], modulation of immune response [13], neuropathic pain [14], the scavenging action on several types of free radicals [15], the frailty in locomotor disease [16] and also anemia [17]. 

2. Authors should modify the conclusion. Not add the text "so it may not be sufficient for final conclusions" (Line 209), but modified the conclusion consistent with the study aim and result.

→We removed the text “so it may not be sufficient for final conclusions”.

The current author's conclusion "The combined administration of iron and TJ-108 is an effective strategy for pregnant women at high risks of PPH such as placenta previa." can not be affirmed because of the small sample number and the authors did not show sufficient direct clinical benefit. How about using the expression "suggested"?

→We revised as followed.

It is suggested that the combined administration of iron and TJ-108 is an effective strategy for pregnant women at high risks of PPH such as placenta previa.

Reviewer 2 Report

I thank the authors thoroughly for addressing the comments of the previous review. 

Author Response

We appreciate your kind advice and thoughtful comments on refining our manuscript.